# Cancerous Tumor Controlled Treatment Using Search Heuristic (GA)-Based Sliding Mode and Synergetic Controller

**DOI:** 10.3390/cancers14174191

**Published:** 2022-08-29

**Authors:** Fazal Subhan, Muhammad Adnan Aziz, Inam Ullah Khan, Muhammad Fayaz, Marcin Wozniak, Jana Shafi, Muhammad Fazal Ijaz

**Affiliations:** 1Department of Electronic Engineering, School of Engineering and Applied Sciences (SEAS), Isra University Islamabad Campus, Islamabad 44000, Pakistan; 2Department of Engineering, King’s College London, London SE1 9NH, UK; 3Department of Computer Science, University of Central Asia, Naryn 722600, Kyrgyzstan; 4Faculty of Applied Mathematics, Silesian University of Technology, 44-100 Gliwice, Poland; 5Department of Computer Science, College of Arts and Science, Prince Sattam bin Abdul Aziz University, Wadi Ad-Dawasir 11991, Saudi Arabia; 6Department of Intelligent Mechatronics Engineering, Sejong University, Seoul 05006, Korea

**Keywords:** nonlinear ordinary coupled differential equation (ncode), Bernstein polynomial (bsp), genetic algorithm (ga), sliding mode controller (smc), synergetic controller (sc), chemotherapy, immunotherapy and optimization

## Abstract

**Simple Summary:**

Cancer is basically a tough condition on a patient’s body where cell grows uncontrollably. Normal cells are affected, which destroys the health of the patient. The main problem in cancer is spreading from one part to another. Therefore, the mathematical modeling of cancerous tumors integrates to check overall stability. A novel approach is introduced such as Bernstein polynomial with combination of genetic algorithm, sliding mode controller, and synergetic control. The proposed solution has easily eliminated cancerous cells within five days using synergetic control. In addition, five cases are incorporated to evaluate error function. In addition, a brief comparative study is added to contrast the simulation results with theoretical modeling.

**Abstract:**

Cancerous tumor cells divide uncontrollably, which results in either tumor or harm to the immune system of the body. Due to the destructive effects of chemotherapy, optimal medications are needed. Therefore, possible treatment methods should be controlled to maintain the constant/continuous dose for affecting the spreading of cancerous tumor cells. Rapid growth of cells is classified into primary and secondary types. In giving a proper response, the immune system plays an important role. This is considered a natural process while fighting against tumors. In recent days, achieving a better method to treat tumors is the prime focus of researchers. Mathematical modeling of tumors uses combined immune, vaccine, and chemotherapies to check performance stability. In this research paper, mathematical modeling is utilized with reference to cancerous tumor growth, the immune system, and normal cells, which are directly affected by the process of chemotherapy. This paper presents novel techniques, which include Bernstein polynomial (BSP) with genetic algorithm (GA), sliding mode controller (SMC), and synergetic control (SC), for giving a possible solution to the cancerous tumor cells (CCs) model. Through GA, random population is generated to evaluate fitness. SMC is used for the continuous exponential dose of chemotherapy to reduce CCs in about forty-five days. In addition, error function consists of five cases that include normal cells (NCs), immune cells (ICs), CCs, and chemotherapy. Furthermore, the drug control process is explained in all the cases. In simulation results, utilizing SC has completely eliminated CCs in nearly five days. The proposed approach reduces CCs as early as possible.

## 1. Introduction

Initially, cancer was considered an untreatable disease. Division and uncontrolled cell growth usually occur because of cancer [1]. Unexpected magnification of cells crosses the limit of a normal level, and cells even migrate to neighbor tissues. However, for cancer cells, mathematical models can be applied for treatment or analysis. Tumor development involves a complicated process. During this approach, when the tumor becomes malignant, the tumor can then spread to the overall body to form secondary tumors [2,3,4]. Globally, cancer is primary cause of death. According to reports from WHO, cancer is the second most dangerous disease in about 112 countries [5]. On the other hand, in 2020, COVID-19 has increased the death rates in comparison to other diseases. Due to current population-based data, the cancer death rate has been reduced since 1990 [6].

The immune system has a direct association in all phases of the tumor lifecycle. Therefore, fast therapy augments the function of a patient’s immune system. This whole process is called cancer immunotherapy, which necessitates work on basic and mathematical computational models to formulate an edge-based silico approach. Apart from clinical methods, immunotherapy and computational models help to innovate this field of study [7]. Immunotherapy is typically used to support the human body’s natural immune system in the battle against cancerous tumors. Initially, CCs dimensions are usually large in size and can be identified with clinical methods. Chemotherapy investigates tumor stability to maintain tumor-free equilibrium by injecting the chemotherapy dose where the drug is and allowing it to mix with the blood. Therefore, the medication is administered into the circulatory system [8]. Achieving optimal procedure of medicines can be utilized to treat cancerous tumors. The main issue is determining the exact dosing plan as well as a proper medication delivery strategy [7].

Many researchers have provided solutions in the field of cancer to facilitate recovery. Computational techniques are considered a possible solution in designing a novel concept in boosting traditional models. The overall paper is based on a new approach to reduce cancerous cells within the body. Sometimes, reduction of cells affects the body in a negative bad way. Therefore, the concept of controllers in the area of cancer is introduced where other techniques such as SMC and SC are also utilized. This paper presents a theoretical comparison with existing techniques and simulation-based approach as well. Many researchers have utilized the basic model of Depillis et al. [9], which is based on traditional therapies. Initially, there was no concept of controllers in reducing the drug rate or eliminating CCs. The theoretical reasoning lies in having information to reduce CCs with respect to a lesser number of days. Controllers such as steepest descent are utilized but can hardly eliminate CCs in eight days [10]. Online recursive calculation [11] has also given the similar results. Therefore, there is a need for more work regarding mathematical models in CCs elimination. 

A solution to the cancer-related problems can likely be determined by establishing mathematical models and understanding their dynamic behaviors. Furthermore, Figure 1 shows the idea of three modes related to cancer, which include immunotherapy, chemotherapy, and SMC and SC as mathematical modeling. Healthy tissue cells consists of immune and host cells that are used in the growth of tumors, which is described in De Pillis [12]. The role of chemotherapy drugs is to have a harmful effect on tumor cells. Thus, a prey–predator model can be used to monitor the growth of tumors within a limited time in immune network [13,14]. Evolutionary computing algorithms are considered the optimal method to address multi-objective engineering problems using spotted hyena optimizer [15]. In addition, for differential evolution, a genetic algorithm can be used to solve the control strategy for cancer treatment drugs [13,14,15,16,17,18,19,20,21,22,23,26]. The main contribution of this research study is as follows:This paper introduces a novel drug that eliminates CCs;Elimination of CCs but also reduction of the effect of chemotherapy on NCs and ICs was also used to bring NCs up to threshold level.A new controller was designed to obtain optimal results where SMC and SC are utilized as drugs;The proposed solution eliminates CCs within five days;Various methods were incorporated to check the performance of the proposed solution with traditional approaches. Further, two basic approaches such as theoretical and simulation were performed to evaluate the results.

The paper’s organization includes Section 2, literature study; Section 3, cancer model with proposed methodology; Section 4, the proposed solution; Section 5, simulation results; Section 6 is comparative discussion, and Section 7 gives the conclusion and future scope.

## 2. Literature Study

This section is about the literature study performed to extract limitation related to cancer using different techniques, which are as follows:

Sima Sarv et al. described the concept of a mathematical model for cancer immunotherapy. A particle swarm optimization (PSO)-based protocol was designed to deal with cellular immunotherapy. However, tumor interaction needs to be better evaluated by using mathematical modeling. A forward-backward approach is considered contemporary but has problems related with time, which led to convergence issues as well [16]. The immune system responds to cancerous tumors. Therefore, to reduce the tumor’s effect on the body overall, immunotherapy is utilized. Due to the human immune system, the fight against cancerous cells is quite easy. In addition, a fixed dose level needs to be deployed to help to reduce CCs. Immunotherapy has attracted researchers with the momentum to utilize antigen T cells, which helps to detect cancerous cells. A special model was designed to reduce cancerous cell growth using chimeric antigen receptor thymus cells (CAR-T cells). Experimentation was performed with in silico tests to select various scenarios. The CAR-T-cell procedure response eliminates cancerous cells and reduces the formation of long-term immuno-memory [17] to maintain the equilibrium that includes cancer cell growth and the immune editing method. Mathematical modeling is quite helpful when it is based on cell population sub-sections. Type 1 interferon receptor (1. IFN) signaling predicts the dominant cancerous cells. For the entire experimentation, triple-negative breast cancer was used [18]. Castrate-resistant prostate cancer (CRPC) with anti-cytotoxic T-lymphocyte-associated protein 4 (anti-CTLA4) was used as a single treatment to estimate results from experimental data. Various constrains were applied to check the performance of CCs, where different drug doses were given to patients to reduce CCs. For better control of CCs, synergy between ipilimumab and sipuleucel-T was utilized [19], giving a possible solution to cancer by using a fractional mathematical model that is based on synergy in between angiogenic and various therapies [20]. Furthermore, a delayed mathematical model of cancerous cells’ immune system is needed to effect drug therapies. There is a relevant, pressing need for a drug-free mechanism that can be understood by the dynamics of a multi-therapeutic approach [21].

Kaouthar Moussa et al. introduced injection scheduling to model cancer treatment, which was used to achieve an optimal level under multiplex tasks. However, interaction can be made possible between CCs and ICs. Many applications normally restructure to schedule injection dose. However, uncertainties need to be further investigated on the initial stage of ICs [22].

Virotherapy improves chemotherapy since the ordinary differential equation (ODE)-based mathematical model balances the interaction among ICs, treatment, and oncolytic cells. This method is useful for completely clearing CCs. Sensitivity examination uses forward techniques to access the effect of virotherapy and chemotherapy. Virus reproduction can be balanced to maintain the tumor equilibrium. Pontryagin’s maximum approach rectifies prediction modeling during continuous treatment of cost and side effects. Furthermore, stability must be investigated to give proper solutions [23]. Table 1 depicts various treatment methods with limitations.

The formation of mathematical model is determined to level up the basic reproduction and stability, which is used to conduct numerical demonstration. An epidemic model of cancer with chemotherapy is a non-linear concept using differential equations. Cancer growth cells with parameters must be constant; therefore, increasing drug dose limits the CCs [24]. Giving a solution for overall orbits and bounded coverage utilizes a phase-space mathematical strategy to limit the CCs growth. Control therapy drives a desirable basin where traditional chemotherapy is not well-applicable [25]. In addition, more constraints regarding the mentioned issue are described in Table 1.

Machine learning and other techniques take too much time in comparison to controllers. High-level data sets are involved to give accurate decisions, whereas training and testing is commonly used and do not give quick solution. The rate of error detection and removal is very tough in machine learning. Due to the mentioned problems, controllers will reduce CCs more easily and effectively. When solving higher-order equations using SMC, order rate reduction occurs. The entire system is highly coupled and non-separable, which is quite hard to solve. SMC and SC swap from easily coupled into de-coupled to reduce disturbance. Overall, a synergetic controller is more reliable than SMC. SMC has a chattering phenomenon, which further leads to low accuracy.

The paper is structured as follows: Section 2 is confined to the tumor model based on a system of coupled differential equations, followed by brief introduction of BSP, GA, and SMC. Section 3 presents the proposed methodology and the design of SMC. Section 4 presents the simulation results and discussion. The conclusion is presented in Section 5.

## 3. Cancer Model with Proposed Methodology

Mathematical models are methods for analyzing the system’s behavior, which gives possible solutions to simulate complex systems [13,18,19,23]. A system of nonlinear coupled ordinary differential equations is discussed in this research study. The proposed solution is based on the below-listed assumptions, which include:CCs and NCs follow logistic growth.ICs and drugs must have natural death rates.NCs have controlled growth, but CCs possess uncontrolled behavior; therefore, population growth will be variable.Drug sources can be either constant or exponential.

### 3.1. Cancer Tumor Model

The cancerous tumor model consists of NCs, CCs, and ICs, where population can be presented by coupled differential equations. Moreover, drug concentration in chemotherapy needs to be monitored using Equation (4). The following Equations (1)–(5) represent NCs, CCs, and ICs with respect to time.
(1)$${\dot{x}}_{1}=a_{2}x_{1}\left(1-d_{2}x_{1}\right)-e_{4}x_{2}x_{1}-r_{3}C\;$$
(2)$${\dot{x}}_{2}=a_{1}x_{2}\left(1-d_{1}x_{2}\right)-e_{2}x_{2}x_{3}-e_{3}x_{2}x_{1}-r_{2}C$$
(3)$${\dot{x}}_{3}=\alpha +\frac{px_{3}x_{2}}{s+x_{2}}-e_{1}x_{3}x_{2}-f_{1}x_{3}-r_{1}C$$
(4)$$\dot{C}=v_{c}\left(t\right)-f_{2}C$$

The initial conditions are
(5)$$\begin{array}{l} x_{1}\left(0\right)=0.9\\ x_{2}\left(0\right)=0.25\\ x_{3}\left(0\right)=0.25 \end{array}$$

The mentioned model describes the metrics of cancer with NCs and ICs. However, $x_{1},\;x_{2},\;$and $x_{3}$ are denoted as NCs, CCs,, and ICs respectively. Furthermore, in Equation (4), C is used for chemotherapy treatment, while the remaining model parameters include$\;r_{1}$, $r_{2}$, and $r_{3}$ coefficients of cell death rate. In addition, $d_{1}$ and $d_{2}$ drugs carry capacities such as $e_{1}\;$to$\;e_{4\;}$. Moreover,$\;f_{1}$ and $f_{2}\;$are considered natural death rates, and $a_{1}\;$and $a_{2}\;$are the growth rates for ICs and drugs, respectively. p is the response rate, and the threshold rate can be symbolized as *s* [6]. The simulation results utilize chemotherapy drugs, and the maximum effect on body cells are observed within 100 days. The obtained results will not reduce the level of NCs, which is $x_{1}\geq 0.75$.

### 3.2. Bernstein Polynomial (BSP)

Approximation functions can be used in BSP to give an optimal solution. Integral and differential equations are used to solve many complex problems. BSP was introduced by Sergi Natanovich in 1912 [15]. However, polynomials with the order *n* and with interval [0, τ] are given in Equations (6)–(12).
(6)$$B_{i,n}\left(t\right)=\left(\begin{array}{c} n\\ i \end{array}\right)\frac{t^{i}{\left(\tau -t\right)}^{n-i}}{\tau^{n}}$$

0≤t≤1, and τ is considering 1.
(7)$$B_{i,n}\left(t\right)=\left(\begin{array}{c} n\\ i \end{array}\right)t^{i}{\left(1-t\right)}^{n-i}$$
(8)$$B_{i,n}\left(t\right)=\{\begin{array}{c} 0\forall i\neq 0\\ 1\forall i=0 \end{array}$$
(9)$$B_{i-1,n-1}\left(1\right)=\{\begin{array}{c} 0\forall i\neq n\\ 1\forall i=n \end{array}$$
(10)$$B_{i,n-1}\left(1\right)=\{\begin{array}{c} 0\forall i\neq n\\ 1\forall i=n-1 \end{array}$$

Lower-ordered polynomials are represented in Equations (11) and (12), which are considered the properties of BSP:(11)$$B_{i,n}\left(t\right)=\left(1-t\right)B_{i,n}\left(t\right)+tB_{i,n}\left(t\right)$$
(12)$$B_{i,n}\left(t\right)=n\left(B_{i-1,n-1}\left(t\right)-B_{i,n-1}\left(t\right)\right)$$

### 3.3. Heuristic Algorithm

GA is the class of nature-inspired heuristic algorithms. The evolutionary computation technique is based on random population of a candidate solution. This is considered the classical method to optimize complex problems by utilizing pairs of chromosomes’ crossover reproduction, mutation, and selection [31]. The genetic algorithm follows the steps below:Random population having unknown length of chromosomes;Candidate solution and mutation are used in genetic algorithm, which is considered the classical method for optimization;Fitness function is utilized to check the desired solution;Crossover, mutation, and selection are found for fitness criteria.

Otherwise, repeat step ii.

### 3.4. Controllers

Controllers are used to give a solution to complex problems, which can be either linear or nonlinear. Usually, the control system regulates undesired responses with uncertainties to the desired reaction. In nonlinear models, integration of linear control systems can be applied. The proposed model is highly nonlinear, which gives the best possible approach for CCs.

### 3.5. Sliding Mode Controllers (SMC)

SMC are used to apply a discontinuous control signal, which works on a state feed-back control mechanism. SMC is a non-linear system, which is used to give stability in two phases. However, defining sliding surface is the first phase, while managing initial states of the system is the second stage. Moreover, when the system reaches the desire state, it is called sliding mode. Complex systems must control finite time while removing parameter variations, order reduction, and decoupling [25,33].

### 3.6. Synergetic Controllers (SC)

SC are used to keep correspondence with nonlinearity and open systems. SC subsystems have dynamic interaction during exchange of information. Nonlinear mathematical models have multi-dimensional properties. However, designing a synergetic model utilizes nonlinear control applications. Presently, SC is a type of dynamic nonlinear system [27,34].

## 4. Proposed Methodology

Cancerous tumor model Equations (1)–(3) are utilized to mimic the error function. For approximation, BSP is demonstrated by using GA, SMC, and SC to the minimize error function of the solution. Linear combination Equations (13)–(16) are evaluated using boundary approaches with different cases.

The below algorithm is considered the proposed solution [24,25,26,27], which consists of BSP, genetic computation, SMC, and SC.
**Algorithm 1** [24,25,26,27]: Model approximation using GA-tuned BSP along with a controller as the proposed drug  **1.** Model approximation using BSP  **2.** Coefficients’ tuning using GA     a. Initialization phase     b. Set parameters for each stage       i. Approximation       ii. Assign number of generation       iii. Generate initial population        1. While         a. Calculate fitness         b. Selection        2. Do         a. Crossover         b. Mutate P(t)        3. End while        4. P(t+1) = New Population  **3.** Applying SMC     a. Set parameters     b. Define sliding surface     c. Design controller to drive initial states to the sliding surface     d. Applying on model     e. Repeat step 1 and 2   **4.** Applying SC     a. Assume macro-variable     b. Design sliding manifold     c. Force the initial states to sliding manifold     d. Repeat step 1 and 2  **5.** Compare SMC and SCStop
(13)$$\begin{array}{l} x_{1}\left(t\right)=\sum_{i=0}^{n}f_{i}B_{i,n}\left(t\right)\\ {\dot{x}}_{1}\left(t\right)=n\left(\sum_{i=1}^{n}f_{i}B_{i-1,n-1}\left(t\right)-\sum_{i=0}^{n-1}f_{i}B_{i,n-1}\left(t\right)\right) \end{array}$$
(14)$$\begin{array}{l} x_{2}\left(t\right)=\sum_{i=0}^{n}g_{i}B_{i,n}\left(t\right)\\ {\dot{x}}_{2}\left(t\right)=n\left(\sum_{i=1}^{n}g_{i}B_{i-1,n-1}\left(t\right)-\sum_{i=0}^{n-1}g_{i}B_{i,n-1}\left(t\right)\right) \end{array}$$
(15)$$\begin{array}{l} x_{2}\left(t\right)=\sum_{n=0}^{n}h_{i}B_{i,n}\left(t\right)\\ {\dot{x}}_{2}\left(t\right)=n\left(\sum_{i=1}^{n}h_{i}B_{i-1,n-1}\left(t\right)-\sum_{i=0}^{n-1}h_{i}B_{i,n-1}\left(t\right)\right) \end{array}$$
(16)$$\begin{array}{l} x_{1}\left(0\right)=\sum_{i=0}^{n}f_{i}B_{i,n}\left(0\right)=f_{0}=0.9\\ x_{2}\left(0\right)=\overset{n}{\underset{i=0}{\sum }}g_{i}B_{i,n}\left(0\right)=g_{0}=0.25\\ x_{3}\left(0\right)=\overset{n}{\underset{i=0}{\sum }}h_{i}B_{i,n}\left(0\right)=h_{0}=0.25 \end{array}$$

However, Equation (16) uses *f_i_*, *g_i_*, and *h_i_*, where (i=1,2, 3, …. n) need to be evaluated through the best possible solution using GA. In addition, $x_{1},\;x_{2},\;\mathrm{and}\;x_{3}$ are initiated in Equations (13)–(16). The unknown constants such as $f_{i},\;g_{i},\;\mathrm{and}\;h_{i}$ easily minimize the objective/error function.

### 4.1. The Error Function

The error function consists of five cases. In them, case-1 contains only NCs and CCs; ICs are involved in case-2, and in case-3, chemotherapy is added. Meanwhile, the rest of the two cases use the concept of elimination of CCs through chemotherapy using SMC and SC. In addition, a drug control process is involved. Different cases are discussed as follows:

#### 4.1.1. Case-1

The first case describes the growth rate of NCs and CCs. Therefore, Equations (17) and (18) explain the mentioned concept. There is no practice involved, such as immunotherapy, chemotherapy, and controllers.
(17)$$Ex_{1}=\frac{1}{11}\sum_{i=0}^{10}({\dot{x}}_{1}(t_{i})-a_{2}x_{1}(t_{i})\left(1-d_{2}x_{1}(t_{i})\right)+e_{4}x_{2}(t_{i})x_{1}(t_{i}){)}^{2}$$
(18)$$Ex_{2}=\frac{1}{11}\sum_{i=0}^{10}({\dot{x}}_{2}(t_{i})-a_{1}x_{2}(t_{i})\left(1-d_{1}x_{2}(t_{i})\right)+e_{3}x_{2}(t_{i})x_{1}(t_{i}){)}^{2}$$

#### 4.1.2. Case-2

Here, immunotherapy is demonstrated where the body’s immune system affects the CCs. However, immunotherapy helps the body defend against CCs. In addition, no concept of chemotherapy and controllers is utilized. Equations (19)–(21) gives the idea of immunotherapy and how it aids in opposing CCs.
(19)$$Ex_{1}=\frac{1}{11}\sum_{i=0}^{10}({\dot{x}}_{1}(t_{i})-a_{2}x_{1}(t_{i})\left(1-d_{2}x_{1}(t_{i})\right)+e_{4}x_{2}(t_{i})x_{1}(t_{i}){)}^{2}$$
(20)$$Ex_{2}=\frac{1}{11}\sum_{i=0}^{10}({\dot{x}}_{2}(t_{i})-a_{1}x_{2}(t_{i})\left(1-d_{1}x_{2}(t_{i})\right)+e_{2}x_{3}(t_{i})x_{2}(t_{i})+e_{3}x_{2}(t_{i})x_{1}(t_{i}){)}^{2}$$
(21)$$Ex_{3}=\frac{1}{11}\sum_{i=0}^{10}({\dot{x}}_{3}(t_{i})-\alpha -\frac{px_{3}(t_{i})x_{2}(t_{i})}{s+x_{2}(t_{i})}+e_{1}x_{3}(t_{i})x_{2}(t_{i})+f_{1}x_{3}(t_{i}){)}^{2}$$

#### 4.1.3. Case-3

As we know, CCs directly affect the process of immunotherapy. However, we added chemotherapy, which tries to reduce CCs. There is no such controller utilized in Equations (22)–(24). However, the main problem is that using only immunotherapy and chemotherapy does not reduce CCs individually. When chemotherapy and immunotherapy are used together, the CCs are reduced. Moreover, chemotherapy disturbs NCs with cancer and also has an effect on ICs.
(22)$$Ex_{1}=\frac{1}{11}\sum_{i=0}^{10}({\dot{x}}_{1}(t_{i})-a_{2}x_{1}(t_{i})\left(1-d_{2}x_{1}(t_{i})\right)+e_{4}x_{2}(t_{i})x_{1}(t_{i})+r_{3}C(t_{i}){)}^{2}$$
(23)$$Ex_{2}=\frac{1}{11}\sum_{i=0}^{10}(\begin{array}{l} {\dot{x}}_{2}(t_{i})-a_{1}x_{2}(t_{i})\left(1-d_{1}x_{2}(t_{i})\right)+e_{2}x_{3}(t_{i})x_{2}(t_{i})+e_{3}x_{2}(t_{i})x_{1}(t_{i})\\ +r_{2}C(t_{i}) \end{array}{)}^{2}$$
(24)$$Ex_{3}=\frac{1}{11}\sum_{i=0}^{10}({\dot{x}}_{3}(t_{i})-\alpha -\frac{px_{3}(t_{i})x_{2}(t_{i})}{s+x_{2}(t_{i})}+e_{1}x_{3}(t_{i})x_{2}(t_{i})+f_{1}x_{3}(t_{i})+r_{1}C(t_{i}){)}^{2}$$

#### 4.1.4. Case-4

In case-4, immunotherapy, chemotherapy, and SMC are used in the investigation. Here, SMC are used to reduce and attempt to quickly eliminate the CCs. Furthermore, a novel error function is designed to speed up the process. The detailed explanation is discussed in Equations (25)–(36).
(25)$$\mu_{x_{2}}\left(t\right)=-\rho_{x_{2}}sgn\left(\sigma_{x_{2}}\right)-\;\partial_{x_{2}}a_{1}x_{2}\left(1-d_{1}x_{2}\right)$$

We added this controller to Equation (2), and after the controller addition ($\mu_{x_{2}}\left(t\right)$), the Equation (2) will be
(26)$${\dot{x}}_{2}=\left(1-\partial_{x_{2}}\right)a_{1}x_{2}\left(1-d_{1}x_{2}\right)-\rho_{x_{2}}sgn\left(\sigma_{x_{2}}\right)-e_{2}x_{3}x_{2}-e_{3}x_{2}x_{1}-r_{2}C$$

$0\leq \partial_{x_{2}}\leq 1\;$is a positive constant and is used for sliding surface. Thus, we define a sliding surface as
(27)$$\sigma_{x_{2}}=m_{1}x_{2}+x_{3}$$

$m_{1\;}\;$is positive. Next, we differentiate Equation (27)
(28)$${\dot{\sigma }}_{x_{2}}=m_{1}{\dot{x}}_{2}+{\dot{x}}_{3}$$

We substitute Equations (26) and (3) in Equation (28), multiplying $\sigma_{x_{2}}\;$on both sides of Equation (28) and following the property$\;\sigma_{x_{2}}sgn\left(\sigma_{x_{2}}\right)=\left|\sigma_{x_{2}}\right|$; thus, the Equation (29) is formed. Describing a term $\eta_{x_{2}}\;$as in Equation (31) and simplifying Equation (30), the Equation (32) will be
(29)$$\sigma_{x_{2}}{\dot{\sigma }}_{x_{2}}=-m_{1}\rho_{x_{2}}\left|\sigma_{x_{2}}\right|+\sigma_{x_{2}}\left(\begin{array}{l} m_{1}\left(\left(1-\partial_{x_{2}}\right)a_{1}x_{2}\left(1-d_{1}x_{2}\right)-e_{2}x_{3}x_{2}-e_{3}x_{2}x_{1}-r_{2}C\right)\\ +\alpha +\frac{px_{3}x_{2}}{s+x_{2}}-e_{1}x_{3}x_{2}-f_{1}x_{3}-r_{1}C \end{array}\right)$$
(30)$$\sigma_{x_{2}}{\dot{\sigma }}_{x_{2}}\leq -\left|\sigma_{x_{2}}\right|\left(m_{1}\rho_{x_{2}}-\left|\begin{array}{l} m_{1}\left(\left(1-\partial_{x_{2}}\right)a_{1}x_{2}\left(1-d_{1}x_{2}\right)-e_{2}x_{3}x_{2}-e_{3}x_{2}x_{1}-r_{2}C\right)\\ +\alpha +\frac{px_{3}x_{2}}{s+x_{2}}-e_{1}x_{3}x_{2}-f_{1}x_{3}-r_{1}C \end{array}\right|\right)$$
(31)$$\eta_{x_{2}}=m_{1}\rho_{x_{2}}-\left|\begin{array}{l} m_{1}\left(\left(1-\partial_{x_{2}}\right)a_{1}x_{2}\left(1-d_{1}x_{2}\right)-e_{2}x_{3}x_{2}-e_{3}x_{2}x_{1}-r_{2}C\right)+\alpha +\frac{px_{3}x_{2}}{s+x_{2}}\;\\ -e_{1}x_{3}x_{2}-f_{1}x_{3}-r_{1}C \end{array}\right|$$
(32)$$\sigma_{x_{2}}{\dot{\sigma }}_{x_{2}}\leq -\left|\sigma_{x_{2}}\right|\eta_{x_{2}}$$
$$\eta_{x_{2}}\geq 0$$

According to the stability of the SMC. Estimated $\rho_{x_{2}}$ from Equation (31) is given in (33) as
(33)$$\rho_{x_{2}}=\frac{\left|\begin{array}{l} m_{1}\left(\left(1-\partial_{x_{2}}\right)a_{1}x_{2}\left(1-d_{1}x_{2}\right)-e_{2}x_{3}x_{2}-e_{3}x_{2}x_{1}-r_{2}C\right)+\alpha +\frac{px_{3}x_{2}}{s+x_{2}}\\ -e_{1}x_{3}x_{2}-f_{1}x_{3}-r_{1}C \end{array}\right|}{m_{1}}+\frac{\eta_{x_{2}}}{m_{1}}$$

Since $-\left|\sigma_{x_{2}}\right|\;\eta_{x_{2}}\;\leq 0\;$by default, the system is therefore asymptotically stable; i.e., $\sigma_{x_{2}}\dot{\sigma_{x_{2}}}\leq 0$. In this case, the use of Equations (1), (3), and (31) results in an error function given by Equations (39)–(42).
(34)$$Ex_{1}=\frac{1}{11}\sum_{i=0}^{10}({\dot{x}}_{1}(t_{i})-a_{2}x_{1}(t_{i})\left(1-d_{2}x_{1}(t_{i})\right)+e_{4}x_{2}(t_{i})x_{1}(t_{i})+r_{3}C(t_{i}){)}^{2}$$
(35)$$E_{x_{2}}=\frac{1}{11}\overset{10}{\underset{j=0}{\sum }}{\left(\begin{array}{l} {\dot{x}}_{2}\left(t_{j}\right)-\left(1-\partial_{x_{2}}\right)a_{1}x_{2}\left(t_{j}\right)\left(1-d_{1}x_{2}\left(t_{j}\right)\right)\\ +\rho_{x_{2}}sgn\left(\sigma_{x_{2}}\right)+e_{2}x_{3}\left(t_{j}\right)x_{2}\left(t_{j}\right)+e_{3}x_{2}\left(t_{j}\right)x_{1}\left(t_{j}\right)+r_{2}C\left(t_{j}\right) \end{array}\right)}^{2}$$
(36)$$Ex_{3}=\frac{1}{11}{\sum_{i=0}^{10}\left({\dot{x}}_{3}(t_{i})-\alpha -\frac{px_{3}(t_{i})x_{2}(t_{i})}{s+x_{2}(t_{i})}+e_{1}x_{3}(t_{i})x_{2}(t_{i})+f_{1}x_{3}(t_{i})+r_{1}C(t_{i})\right)}^{2}$$

#### 4.1.5. Case-5

Case-5 is just like the previous experiment but with immunotherapy and chemotherapy, and an updated SC is utilized. Due to this method, CCs are reduced very quickly. From Equations (37)–(53), the mentioned detailed experimentation was performed.

In the case of the controller, the Equation (2) will be
(37)$${\dot{x}}_{2}=a_{1}x_{2}\left(1-d_{1}x_{2}\right)-e_{2}x_{2}x_{3}-e_{3}x_{2}x_{1}-r_{2}C+\mu_{x_{2}}$$

Here, $\mu_{x_{2}}$ is a controller
(38)$$\psi =f(x_{2})$$
(39)$$\psi =m_{2}(x_{2}-x_{2r})$$

ψ is a macro variable, and $m_{2}$ is a positive constant, while the $x_{2r}=0$ is the reference of CCs
(40)$$\psi =m_{2}x_{2}$$

We differentiate with respect to time, *t*
(41)$$\dot{\psi }=m_{2}{\dot{x}}_{2}$$
and we define a manifold
(42)$$\dot{\psi }+\frac{\psi }{\tau }=0$$

Next, we substitute Equations (39) and (40) in Equation (41)
(43)$$m_{2}({\dot{x}}_{2}+\frac{x_{2}}{\tau })=0$$

We substitute Equation (37) in Equation (43), and with some manipulation, we obtain
(44)$$\mu_{x_{2}}=-a_{1}x_{2}\left(1-d_{1}x_{2}\right)+e_{2}x_{2}x_{3}+e_{3}x_{2}x_{1}+r_{2}C\;\;-\frac{x_{2}}{\tau }$$

Now, we substitute controller Equation (44) in Equation (37), and after simplification, we obtain
(45)$${\dot{x}}_{2}=-\left(\frac{1}{\tau }\right)x_{2}$$

The solution of Equation (45) is
(46)$$x_{2}=x_{2}(0)e^{-\frac{t}{\tau }}\;x_{2}{\left(t\right)}_{t\to \infty }=0$$

We use the Lyapunov function to check the stability of the controller
(47)$$L=\frac{1}{2}\psi^{2}$$
(48)$$\dot{L}=\dot{\psi }\psi$$

We transfer the value of $\dot{\psi }$ from Equation (42) in Equation (48); we arrive at
(49)$$\dot{L}=\frac{-\psi^{2}}{\tau }$$
(50)$$L=L(0)e^{\frac{-2t}{\tau }}\;$$

t→∞; the system approaches zero, so the model is asymptotically stable.
(51)$$Ex_{1}=\frac{1}{11}\sum_{i=0}^{10}({\dot{x}}_{1}(t_{i})-a_{2}x_{1}(t_{i})\left(1-d_{2}x_{1}(t_{i})\right)+e_{4}x_{2}(t_{i})x_{1}(t_{i})+r_{3}C(t_{i}){)}^{2}$$(52)$$x_{2}=x_{2}(0)e^{-\frac{t}{\tau }}$$(53)$$Ex_{3}=\frac{1}{11}{\sum_{i=0}^{10}\left({\dot{x}}_{3}(t_{i})-\alpha -\frac{px_{3}(t_{i})x_{2}(t_{i})}{s+x_{2}(t_{i})}+r_{1}C(t_{i})+e_{1}x_{3}(t_{i})x_{2}(t_{i})+f_{1}x_{3}(t_{i})\right)}^{2}\;$$

In all the above cases, the error function to be minimized is as follows:(54)$$E_{optimal}=\mathrm{minimum}\left(E_{N}+E_{T}+E_{I}\right)$$

In this section, Table 2 represents different controllers using metric values that vary from either 0 to 1. Therefore, the estimated values and reduction of CCs are incorporated with SMC and SC.

## 5. Numerical Results and Discussion

Previously, two methods were utilized in the literature, which are constant, continuous and pulsed chemotherapy. While trying to reduce cancerous cell using therapies, logic is crucial. To eliminate cancerous cells completely from body, a better approach than using medicines is required. The above figure presents the level of chemotherapy with constant and continuous dose methods. Initially, chemotherapy starts from zero, and then after some time reaches the maximum. On the other hand, the constant approach uses fixed doses throughout the chemotherapy.

Figure 2 depicts the behavior of constant and continuous chemotherapy drugs that are given to a patient with the passage of time. An exponential dose becomes reduced and might be eliminated, while a constant dose is applied regularly. The average value of a constant dose is calculated to be about 0.9942. However, a continuous dose is approximately equal to 0.7499.

Figure 3 shows the NCs were reduced and died down, while CCs increased, which is not an optimal case for the body. This figure depicts that no controller or treatment was used; there was only the interaction between normal and cancerous cells, in which NCs were reduced in levels due to cancer. However, CCs increased from their level in comparison with normal cells. During experimentation, the initial value of CCs was 0.25. Therefore, 0.25 is considered the threshold for cancer patients. If the value of CCs is increased from 0.25, then the patient will die on the spot.

Figure 4 illustrates the concept of case-2, where immunotherapy was added, which reduced the NCs and CCs. Moreover, ICs were in the rising phase, which is shown in case-2. Separately, immunotherapy is not a very appropriate method. In Figure 4, there is clear indication that CCs’ growth slowed down but still increased with the passage of time. More interestingly, the result shows that there is a need for other treatments as well with immunotherapy or controllers.

In Figure 5, case-3a represents that chemotherapy completely eliminated CCs after one hundred days; there was no such controller utilized, but the dose was constant. In Figure 6, case-3b describes the continuous dose with chemotherapy, where CCs were eliminated within eighty days. However, NCs and ICs became disturbed, which is not good for the body. In Figure 7, case-4a is illustrated, in which SMC were applied, causing CCs to reach the minimum level. Apart from that, ICs and NCs were not disturbed.

Moreover, in Figure 8, case-4b represents that CCs reduced in about sixty days. Due to the speedy behavior of chemotherapy, CCs and NCs reached the very minimum threshold, which is quite dangerous for the body. In case-5a, whose results are shown in Figure 9, the contemporary SC was applied with chemotherapy at a constant dose. Optimal results were obtained in which NCs and ICs were at maximum level. However, CCs were reduced to level zero within five days. Figure 10 is the result of case-5b, which depicts that using SC, CCs were removed during five days with continuous chemotherapy.

In Figure 5 and Figure 6, the treatment utilized was the same and was based on chemotherapy. However, Figure 5 shows a fixed-dose treatment. In Figure 6, on the other hand, exponential-dose chemotherapy was used. Further, normal cells were not disturbed too much in Figure 5. Immune cells were at the maximum level, but CCs took a long time to be eliminated from the patient’s body.

Figure 6 shows completely different results in comparison with Figure 5. Normal and immune cells showed negative variation. CCs were reduced quickly, in contract with Figure 5.

However, Figure 7 shows results using the SMC controller, whereas in Figure 5 and Figure 6, no such controller was used. A fixed dose of chemotherapy was utilized, which is commonly called constant as well. Comparing Figure 7 and Figure 8, the controller utilized is the same, but Figure 7 shows the optimal results. In addition, normal and immune cells were not disturbed in Figure 7, which is a positive sign for the patient.

It is quite clear in Figure 7 that normal and immune cells show better results. Meanwhile, Figure 8 also utilizes an SMC controller, but CCs were reduced more quickly than in Figure 7. Therefore, Figure 8 shows the main objective was achieved: CCs were eliminated in 50 to 60 days.

However, Figure 9 and Figure 10 both show use of the synergetic controller, which is a completely different approach than SMC, while, Figure 9 presents constant or fixed-dose chemotherapy. Working with the SC approach, ICs and NCs were not affected. In addition, as mentioned, CCs were removed within 5 days. Thus, SC is the only approach that gives better results.

Figure 10 illustrates the same results of CCs reduction in five days, similar to Figure 9. However, in Figure 10, there is a clear negative variation in NCs and ICs using the synergetic controller. In addition, more interestingly, in Figure 10, continuous-dose chemotherapy was utilized.

Figure 11 shows a detailed comparison of CCs with SMC, CCs with SC, and NCs and ICs. The overall results of Figure 11 are based on fixed-dose chemotherapy, while results are quite preferable where CCs are completely eliminated within five days using a synergetic controller. Coupled differential equations are used, and therefore, CCs are shown to have an effect on ICs and NCs using a synergetic controller.

Overall, discussion of Figure 12 is presented in the earlier figures, where chemotherapy with continuous doses of SMC and SC were used. Therefore, in Figure 12, CCs were reduced earlier, but was negative variation normal and immune cells.

In Figure 11, case-6a shows the comparison of SMC and SC, in which we utilized constant-dose chemotherapy, and ICs and NCs are at normal level. However, CCs were eliminated in five days with the help of SC. However, using SMC, CCs were minimized in about eighty days. Moreover, in Figure 12, case-6b illustrates an exponential continuous dose with chemotherapy, where using SMC, CCs are reduced nearly in sixty days. However, NCs and ICs are disturbed, which affects the body. In contract with SMC, SC completely removed CCs within five consecutive days. In Figure 13, case-6c shows the idea that SC especially is designed for CCs; later, we utilize the same concept in all other equations. In the above graph, ICs and NCs are at a normal level, while, SMC for CCs showed worse results because CCs survived for about eighty days. The chemotherapy dose is constant in Figure 13. In addition, ICs are at maximum level, which is about 0.25. Furthermore, NCs are near to 0.9, which means NCs are not reduced. This case is basically considered optimal for patients. In Figure 14, case-6d shows similar results as with SC but with SMC, as CCs are reduced in about sixty days using chemotherapy at a continuous dose.

In Figure 15, Ta is shown where different values are used for SC to remove CCs. Therefore, when Ta = 0.01, CCs are eliminated within five days although if Ta = 0.04, CCs are reduced in about twenty days. In addition, for various values like 0.07, 0.1, and 0.2, SC was evaluated to reduce CCs, which is presented in Figure 14.

To reduce the effect of cancerous tumors, chemotherapy and immunotherapy can be utilized. Additionally, hybrid therapies use various types of controllers such as synergetic and sliding mode controllers. Therefore, these controllers can be used as drug treatments to optimize cells of the body. Synergetic controllers are more efficient in comparison with other techniques.

## 6. Comparative Discussion

Table 3 depicts the comparison of various treatments and controllers used in the study. The proposed approach shows better results in comparison with traditional techniques. Moreover, Depillis et al. [20] demonstrated improved NC levels in contrast with CCs and ICs. However, due to this method, CCs are not eliminated properly. On the other hand, Omar et al. [14] implemented a multi-objective swarm model where NCs cannot exceed the minimum threshold. Apart from that, chemo-immunotherapy with SMC was utilized, which destroyed the CCs in about forty-five days [15], while, in the proposed solution, SC reduced CCs within five days.

However, mathematical modeling for chemotherapy and immunotherapy is rarely used. Normally, immunotherapy is utilized to boost the immune system of the body. The main aim of therapies is to reduce the effect of cancer cells; to target cancerous cells, immunotherapy is quite effective. In comparison with other therapies, mathematical models of cytotoxic chemotherapy were utilized by depillis et al. [9] to eliminate CCs. Formulating a novel chemotherapeutic protocol that improves defense strategies against cancer cells requires a brief understanding of immune system. Therefore, mathematical models of immunotherapies utilize a complex network of cells. Traditional chemotherapies have been studied but without the role of controller. Without the use of a controller, CCs are hardly reduced in about seventy days.

In reference [10], a controller-based model was designed to find the optimal rate of cancer drugs. During therapies, the drug rate is a major factor that can reduce cancer cells. However, due to excess use of the drug, sometimes healthy cells within the body can also experience reduced levels of growth. Therefore, steepest descent technique is utilized to give logical reasoning to improve adaptive controllers. The online recursive calculation approach is used to check the performance of metrics. In the results, NCs improves in a slow way, but CCs are still reduced in about eighty days, which is, again, an alarming condition.

Samira et al. [11] tried to resolve issues related with drug rate and the time needed for giving drugs during immunotherapy. The similar depillis model was implemented by applying the theory of optimal impulsive method, where five differential equations are elaborated with cancer and immune cells. In this study also, CCs were eliminated in about one hundred days, which are not presently better results in comparison with reference [20,28].

Omar et al. [27] presented the concept of combining an optimal control theory with swarm intelligence techniques. Here, in this study, the main focus was drug concentration, where the hybrid approach was far better than other algorithms. To verify the performance, second-order coefficient was used with a multi-objective approach. According to this technique, CCs were easily reduced within fifty days, which presents better results than the above-mentioned study. However, in [35], a new approach of mathematical modeling of CAR-T immunotherapy eliminated cancer cells similarly within fifty days.

Minimizing CCs while injecting drug formulations of Pontryagin’s maximum principle established a better balance with cost effectiveness of the control variables. Das et al. [30] eliminated CCs using a quadratic control mechanism in about forty days. However, using an SMC controller, CCs died in forty-five days [28]. Therefore, both cases’ results were not optimal and can be further formulated in the near future.

Dehingia et al. [36] recently introduced a technique used to understand the optimality of immune chemotherapy. Feasible domains of various mathematical models are validated using the condition of equilibrium points. This process is used to solve drug toxicity during immune chemotherapy. Further, through this quadratic methodology, CCs were easily reduced in twenty days.

For dealing with cancerous cells, a novel concept of SC was designed in this study. GA, SMC, and SC mathematical models were utilized as a hybrid combination that completely eliminates CCs within five days. This study is compared with existing techniques in the simulation results, where the proposed approach presents superiority. In addition, the theoretical analysis gives a brief overview to compare the proposed solution with previous studies. Therefore, due to both methods, namely simulation and theoretical approach, this study depicts the optimal results of the proposed approach.

This study is limited to the analysis of cancerous tumors and their controlled treatment in the domain of mathematical models at present. Clinical validation of the proposed treatment protocol can be investigated as a prospect study subject to the realization of drugs imitating the effects of SC and SMC utilized in this study. 

## 7. Conclusions

Mathematical models are utilized to evaluate the complex behavior of CCs and NCs, where immune cells are reduced in number due to the fast growth of CCs. Overall, drug dosages need to be exponential with the passage of time. Cancer is considered one of the leading diseases, which arise from uncontrolled division of NCs into CCs. Cancer can be directly reduced or eliminated if CCs can be detected early. For improving the life of cancer patients, various treatment methods such as chemotherapy, immunotherapy, or mathematical modeling need to be utilized for early detection of CCs. This research study consists of using GA, SMC, and SC to reduce the effect and eliminate CCs as soon as possible. However, the proposed work is compared with the existing models to evaluate its performance. The SC easily reduces the CCs in nearly five days and maintains the patient’s health state as well. NCs and ICs are improved by using SMC and SC, which is considered an optimal approach for elimination of CCs. SC was determined as the best possible approach as an anti-tumor drug. Figure 13 shows he best optimal result for CCs elimination and also in keeping NCs and ICs at their maximum levels using constant-dose chemotherapy along with SC. In the previous three to four decades, cancer prevention has moved from medicinal studies such as immunotherapy and chemotherapy to mathematical modeling. However, in the future, various evolutionary computational techniques such as ant colony optimization [38], particle swarm optimization [39], differential evolution, and artificial bee colony along with different controllers can be investigated. Additionally machine learning, deep learning, and stochastic Markov chain distribution [40] will envision mathematical modeling not only for CCs but also for different diseases. Further, image classification, data-driven classification models, disease detection, feature classification, and blood vessel segmentation for CCs can be utilized to give possible solutions in the near future [37,38,39,40,41,42,43].

## Figures and Tables

**Figure 1 cancers-14-04191-f001:**
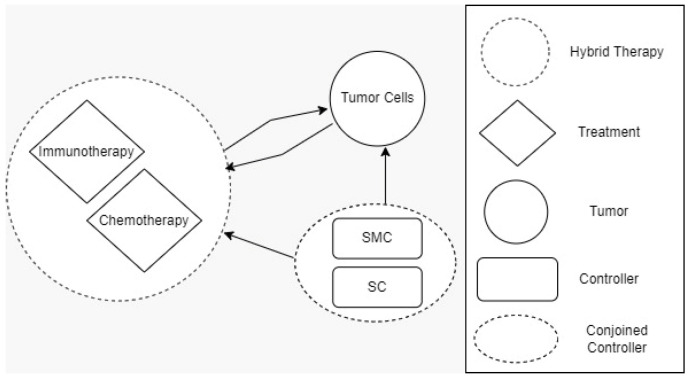
Three different modes for cancer-related problems.

**Figure 2 cancers-14-04191-f002:**
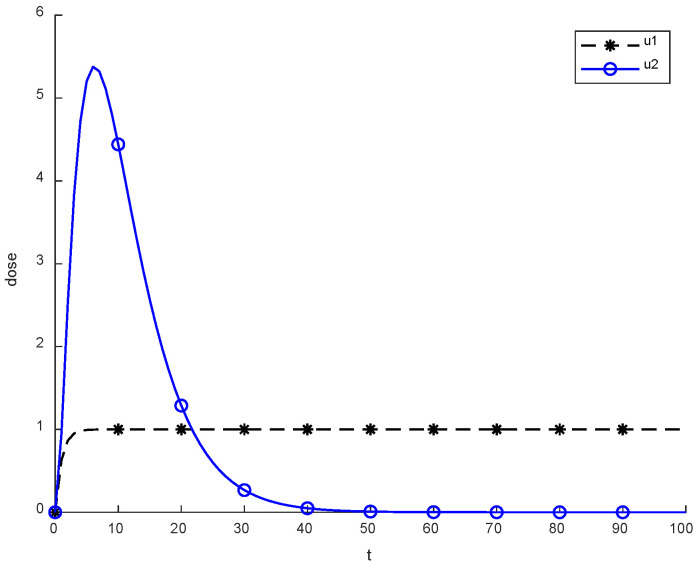
Behavior of constant and continuous chemotherapy.

**Figure 3 cancers-14-04191-f003:**
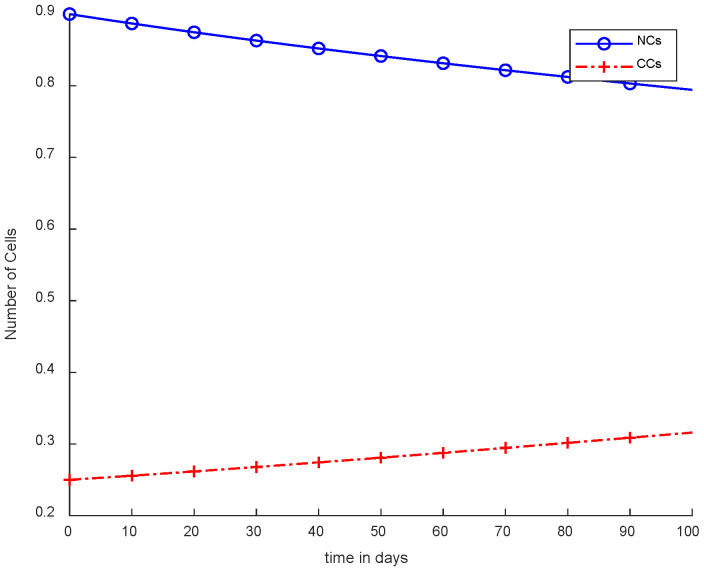
Without ICs, chemotherapy, and controller.

**Figure 4 cancers-14-04191-f004:**
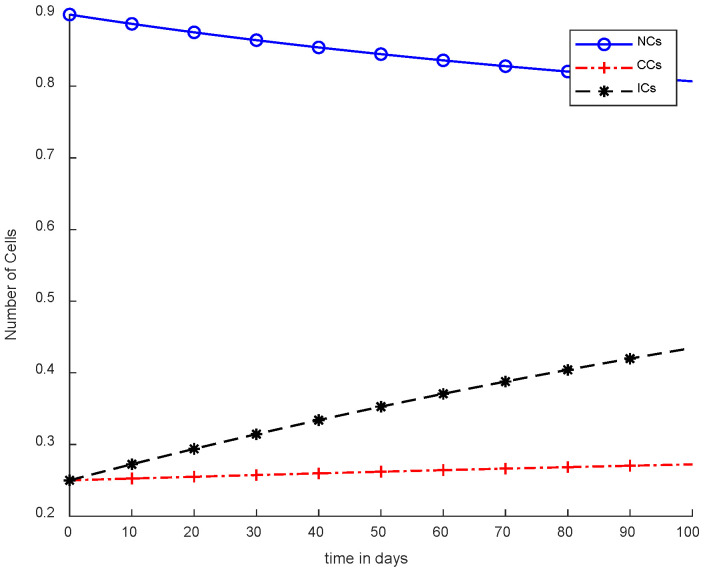
Without chemotherapy and controllers.

**Figure 5 cancers-14-04191-f005:**
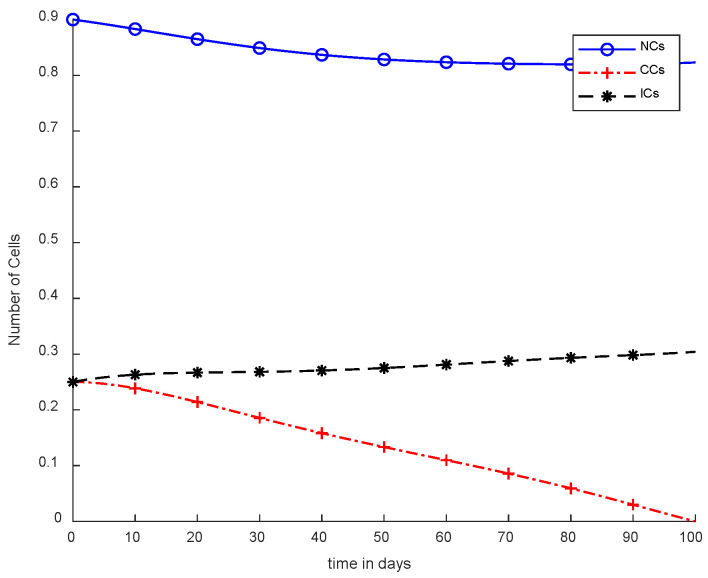
With chemotherapy at a constant dose and without controller.

**Figure 6 cancers-14-04191-f006:**
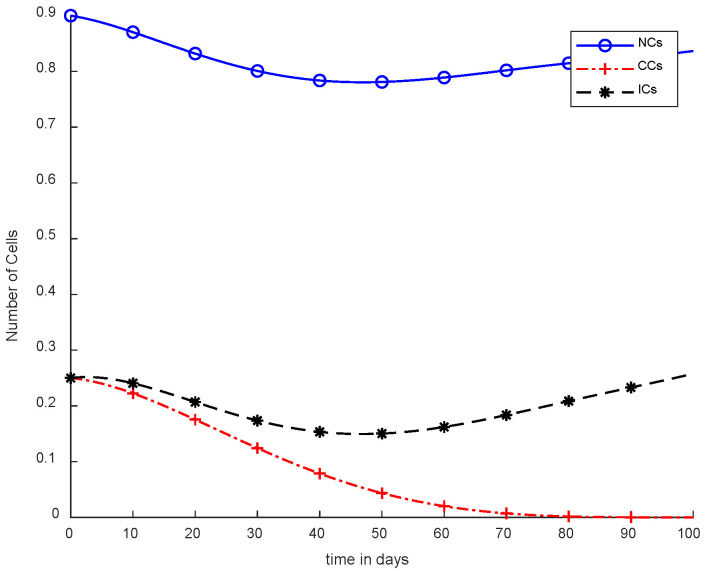
With chemotherapy at a continuous dose and without controller.

**Figure 7 cancers-14-04191-f007:**
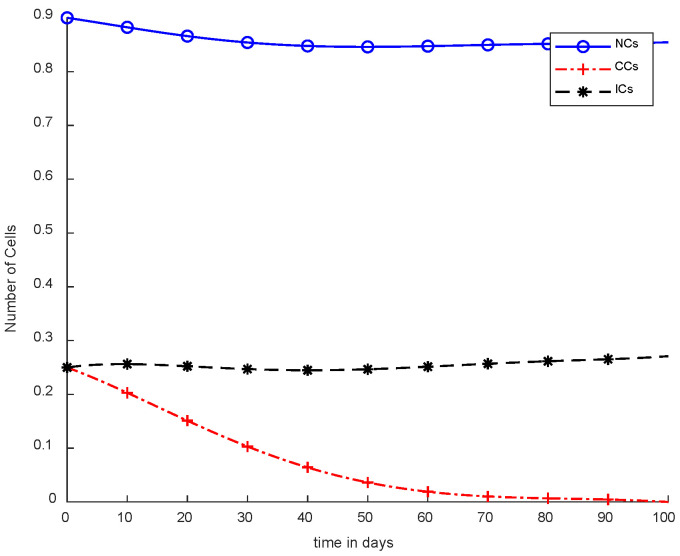
With chemotherapy constant dose and SMC for CCs Killer.

**Figure 8 cancers-14-04191-f008:**
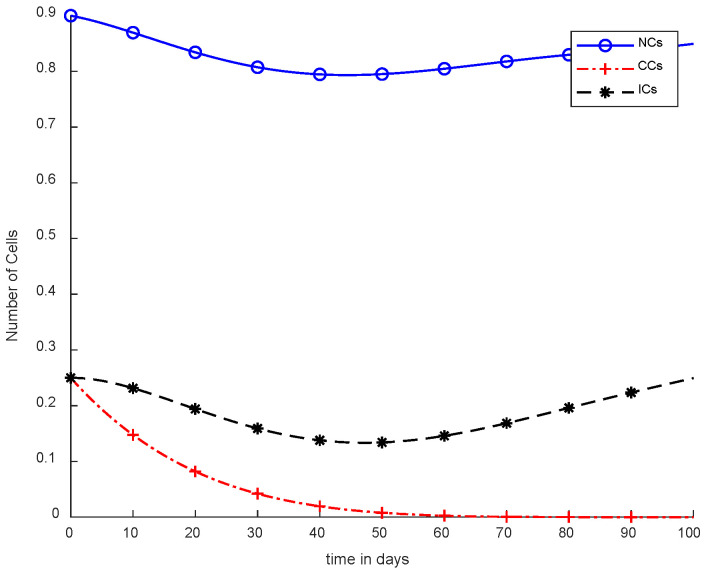
With chemotherapy at a continuous dose and SMC to kill CCs.

**Figure 9 cancers-14-04191-f009:**
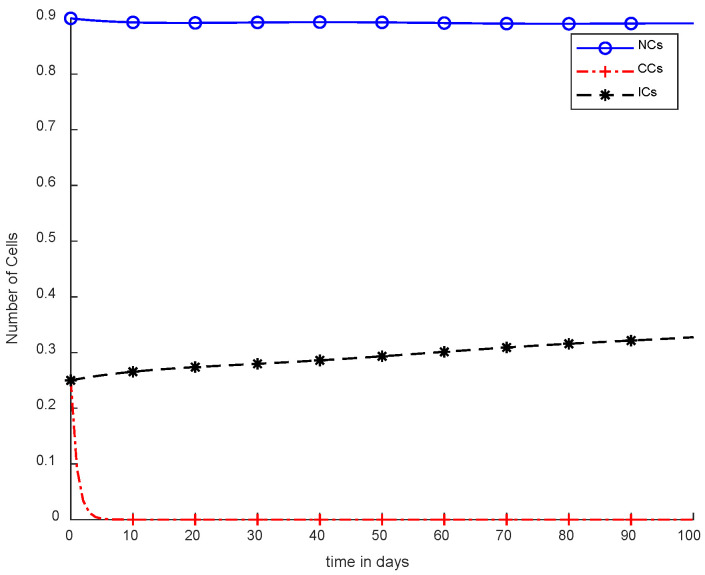
With chemotherapy at a constant dose and SC to kill CCs.

**Figure 10 cancers-14-04191-f010:**
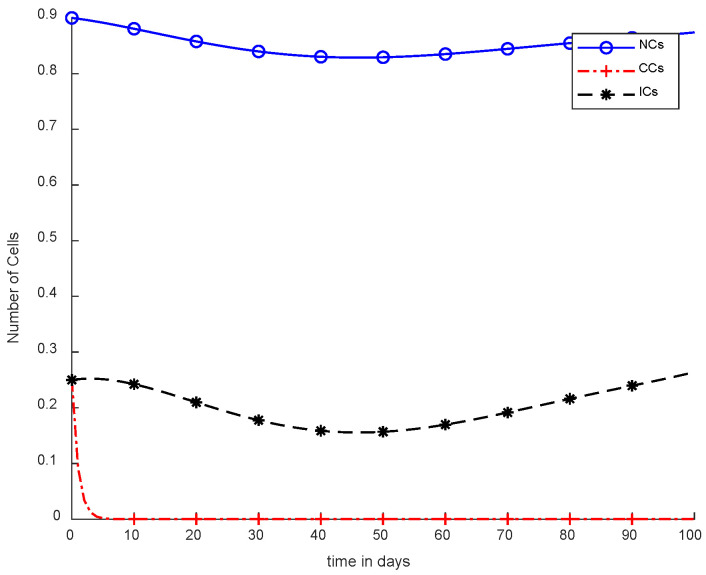
With chemotherapy at a continuous dose and SC to kill CCs.

**Figure 11 cancers-14-04191-f011:**
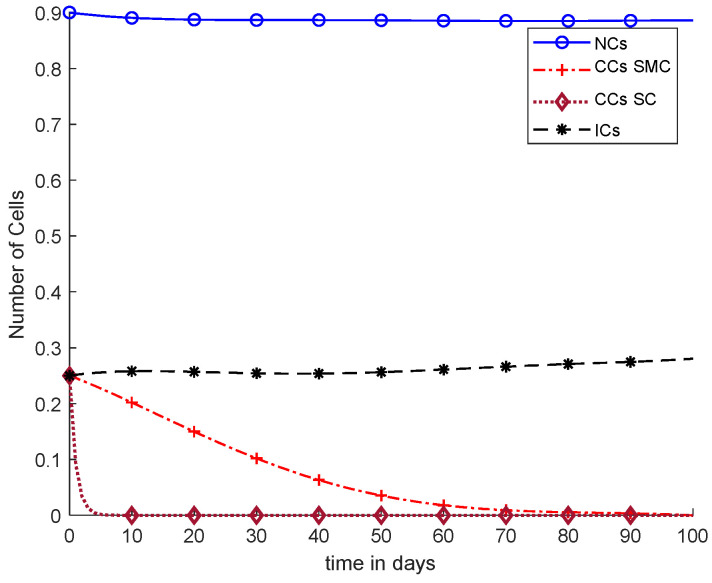
With chemotherapy at a constant dose, SMC on CCs (‘+’ line) with effect on all equations, and SC on CCs (‘−’ line).

**Figure 12 cancers-14-04191-f012:**
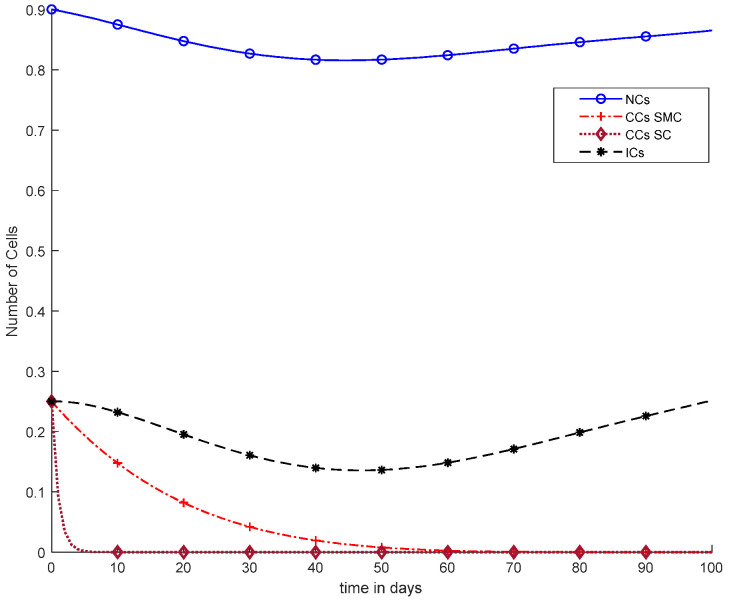
With chemotherapy at a continuous dose and SMC on CCs (‘+’ line) with effect on all equations and SC on CCs (‘−’ line).

**Figure 13 cancers-14-04191-f013:**
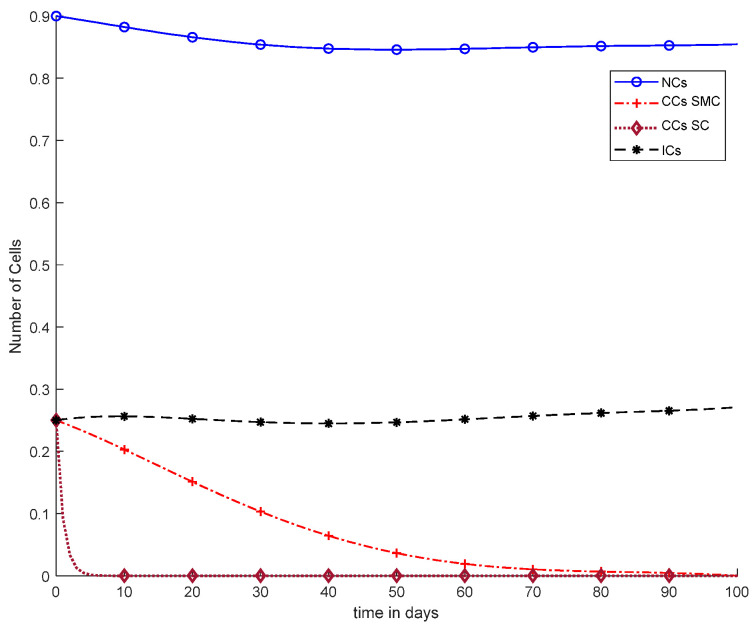
With chemotherapy at a constant dose, SMC on CCs (‘+’ line), and SC on CCs (‘−’ line) with effect on all equations.

**Figure 14 cancers-14-04191-f014:**
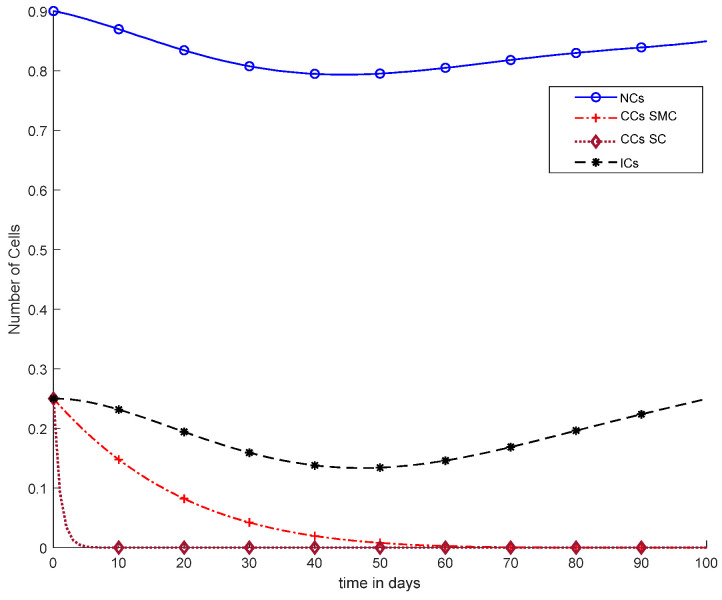
With chemotherapy at a continuous dose, SMC on CCs (‘+’ line), and SC to kill CCs (‘−’ line) with effect on all equations.

**Figure 15 cancers-14-04191-f015:**
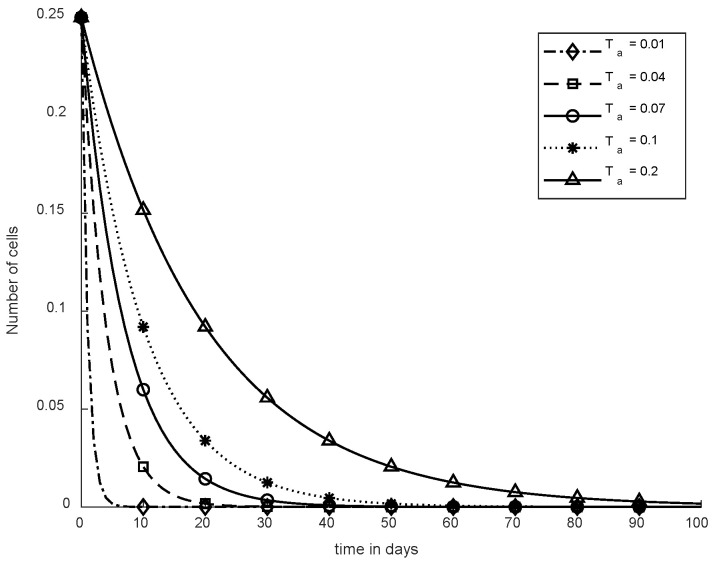
Convergence time of SC.

**Table 1 cancers-14-04191-t001:** Different treatment methods with limitations.

Treatment and Controller	Behavior	Limitations
Pulsed chemotherapy protocol [9]	Oscillatory behavior of CCs and ICs	CCs not removed completely
Direct collocation as an optimal control with continuous chemotherapy [19]	Oscillation in ICs, slow reduction of CCs	CCs eliminated within 70 days, NCs reduced to dangerous level
Traditional pulse chemotherapy [20]	Reduction of CCs and NCs	CCs still remaining, NCs die down to minimum threshold
Optimal control with chemotherapy [20]	CCs slowly removed	Elimination of CCs within 70 days
Chemo-immunotherapy with optimal control [20]	Oscillatory behavior of NCs and ICs	Treatment destroys the CCs, NCs, and ICs
Multi-objective swarm as an optimal control with chemotherapy [14]	Nonlinear behavior of treatment, NCs and CCs.	NCs reduced to minimum edge, so for the time being, treatment is stopped to recover NCs to a safe level.
Chemo-immunotherapy of triple-negative breast cancer [29]	ICs remain at very low level	CCs eliminated after 60 days
Optimal administration protocols for immunotherapies [22]	Nonlinear behavior of CCs elimination	CCs eliminated after 40 days
Chemo-immunotherapy with SMC [15]	CCs eliminated from the patient’s body within 45 days.	The CCs elimination is good but can be enhanced.

**Table 2 cancers-14-04191-t002:** Different controllers using parameters with values.

Parameters	Values	Estimated	Description
$\partial_{x_{2}}$	1	0 to 1	Reduction coefficient of growth rate of CCs
$\eta_{x_{2}}$	0	0 to 0.8	Positive constant
$\rho_{x_{2}}$	0	0 to 1	Coefficient of controller nonlinear term
$\tau_{a}$	0.01	0.01 to 0.2	Convergence time of SC
$m_{1}$	1	1	Coefficient of SMC
$m_{2}$	1	0 to 1	Coefficient of SMC

**Table 3 cancers-14-04191-t003:** Comparative study.

Treatment and Controller	Cells	Description
Traditional pulsed chemotherapy without controller [9]	NCs	NCs reduced to minimum level.
	CCs	CCs held at maximum level.
	ICs	Little increase in ICs was observed.
Chemotherapy with optimal control [9]	NCs	NCs hit minimum level and when treatment halted rose to maximum level.
	CCs	Approximately, in 70 days, CCs fell to zero.
	ICs	ICs also increased to a good level.
Chemotherapy and angiotherapy along with adaptive controller [10]	NCs	NCs very slowly increased to a healthy state.
	CCs	More than 80 days needed to decrease to minimum level.
	ECs	During treatment, ECs increased and after that decreased
.Multi immunotherapy [11]	CCs	CCs reduced to minimum level within 100 days but were not completely removed.
	ICs	Also decreased.
Multi objective swarm with optimal control [27]	NCs	When NCs reached minimum threshold, treatment was stopped for a short time for the recovery of NCs.
	CCs	Approximately, in 50 days, CCs fell to zero.
	ICs	ICs increased to a good level.
Chemo-immunotherapy along with SMC controller [15]	NCs	NCs held at maximum level.
	CCs	CCs eliminated within 45 days.
	ICs	ICs achieved a good level.
Multi Chemo-immunotherapy along with Quadratic control [35]	NCs	NCs increased after CCs elimination.
	CCs	CCs eliminated approximately in 40 days.
	ICs	ICs also increased slightly after CCs elimination.
Chemo-immunotherapy along with Quadratic control [28]	CCs	CCs exterminated approximately in 20 days.
	ICs	ICs rose to maximum level after 100 days.
Optimal administration protocols for cancer immunotherapies [36]	CCs	CCs eliminated approximately at 35 to 40 days.
	ICs	ICs also rose after CCs elimination.
Mathematical modelling of CAR-T immunotherapy [32]	CCs	CCs eliminated approximately within 50 days.
	ICs	ICs increased after CCs elimination.
Mathematical modelling of Chemo-immunotherapy in Triple-Negative Breast cancer [21]	CCs	CCs completely removed within 60 days.
	ICs	ICs achieved maximum level after CCs elimination.
Chemo-immunotherapy along with conjoined SMC and SC controller (proposed)	NCs	NCs held to maximum level.
	CCs	CCs eliminated within 5 days.
	ICs	ICs also held to maximum level

## Data Availability

The data presented in this study is available on request from the corresponding author.

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
