# Peer review of "Cancerous Tumor Controlled Treatment Using Search Heuristic (GA)-Based Sliding Mode and Synergetic Controller"

_cancers, 2022, doi:10.3390/cancers14174191_

Round 1

Reviewer 1 Report

The article entitled “Cancerous Tumor Controlled Treatment Using Search Heuristic (GA) Based Sliding Mode & Synergetic Controller” is well-written and, from my point of view, would be of interest for the readers of Cancers. In spite of this and before its publication I would suggest authors to perform the changes described below:

Line 32: the meaning of NC, IC, CC is not included in the abstract. Please, include it.

Please review the whole manuscript in search of some sentences that are difficult to understand in English. Please see below three examples:

Line 41 “Cancer is considered significant threat to life”

Lines 47-48 “According to the reports of WHO, cancer is 47 the second leading disease in about 112 countries”

Line 63: “There exists variety of solutions by many practitioners in the field of cancer.”

Please note that the first time an acronym or abbreviation is employed must be described. Check the whole text and arrange such issue. And example of this can be found in line 57CCs.

Section 3.1. Cancer Tumor Model please, check the description of each variable and modify in order to make it more easy to undestand.

In line 207 it is said “Else repeat step-ii”. What is step-ii.

Author Response

Dear Reviewer

Kindly check the attached file. 
Best
Authors

Reviewer 2 Report

This manuscript, written by Dr Subhan, original research, with the title of "Cancerous Tumor Controlled Treatment Using Search Heuristic (GA) Based Sliding Mode & Synergetic Controller" showed different methods for matemathical modelling of cancer growth and the relationship with immune cells and normal cells. This research consisted on genetic algorithm, Sliding Mode Controller, and Synergetic Controller (SC). The authors concluded that in the future mathematical modelling will not only be applied to cancer cells but also to other diseases.

Comments:

(1) As I understand, the authors are showing theoretical results. I wonder how these models correlate with real experimental data.

(2) Could you please explain what are the practical applications of this research?

(3) Could you please add legends in the figures explaining how to interpret the displayed data? Most of the readers of this journal will be medical doctors or medical researchers, and I have serious doubts that they can understand the mathematics, nor interpret the results that are being shown.

(4) Regarding tumor initiation and progression, do the authors favor a linear model, o oncotree model, DAG, or evolutionaly dynamics?

Author Response

(The authors gave the same response as above.)

Round 2

Reviewer 1 Report

After the changes performed by the authors, I consider that the manuscript is now ready for its publication. Congratulations.

Author Response

Dear Reviewer

Thanks for the positive feedback.

Best

Authors